# Conditional Swap Regret and Conditional Correlated Equilibrium

**Mehryar Mohri**
Courant Institute and Google
251 Mercer Street
New York, NY 10012
mohri@cims.nyu.edu

**Scott Yang**
Courant Institute
251 Mercer Street
New York, NY 10012
yangs@cims.nyu.edu

## Abstract

We introduce a natural extension of the notion of swap regret, *conditional swap regret*, that allows for action modifications conditioned on the player's action history. We prove a series of new results for conditional swap regret minimization. We present algorithms for minimizing conditional swap regret with bounded conditioning history. We further extend these results to the case where conditional swaps are considered only for a subset of actions. We also define a new notion of equilibrium, *conditional correlated equilibrium*, that is tightly connected to the notion of conditional swap regret: when all players follow conditional swap regret minimization strategies, then the empirical distribution approaches this equilibrium. Finally, we extend our results to the multi-armed bandit scenario.

## 1 Introduction

On-line learning has received much attention in recent years. In contrast to the standard batch framework, the online learning scenario requires no distributional assumption. It can be described in terms of sequential prediction with expert advice [13] or formulated as a repeated two-player game between a player (the algorithm) and an opponent with an unknown strategy [7]: at each time step, the algorithm probabilistically selects an action, the opponent chooses the losses assigned to each action, and the algorithm incurs the loss corresponding to the action it selected.

The standard measure of the quality of an online algorithm is its *regret*, which is the difference between the cumulative loss it incurs after some number of rounds and that of an alternative policy. The cumulative loss can be compared to that of the single best action in retrospect [13] (*external regret*), to the loss incurred by changing every occurrence of a specific action to another [9] (*internal regret*), or, more generally, to the loss of action sequences obtained by mapping each action to some other action [4] (*swap regret*). Swap regret, in particular, accounts for situations where the algorithm could have reduced its loss by swapping every instance of one action with another (e.g. every time the player bought Microsoft, he should have bought IBM).

There are many algorithms for minimizing external regret [7], such as, for example, the randomized weighted-majority algorithm of [13]. It was also shown in [4] and [15] that there exist algorithms for minimizing internal and swap regret. These regret minimization techniques have been shown to be useful for approximating game-theoretic equilibria: external regret algorithms for Nash equilibria and swap regret algorithms for correlated equilibria [14].

By definition, swap regret compares a player's action sequence against all possible modifications at each round, independently of the previous time steps. In this paper, we introduce a natural extension of swap regret, *conditional swap regret*, that allows for action modifications conditioned on the player's action history. Our definition depends on the number of past time steps we condition upon.

As a motivating example, let us limit this history to just the previous one time step, and suppose we design an online algorithm for the purpose of investing, where one of our actions is to buy bonds and another to buy stocks. Since bond and stock prices are known to be negatively correlated, we should always be wary of buying one immediately after the other – unless our objective was to pay for transaction costs without actually modifying our portfolio! However, this does not mean that we should avoid purchasing one or both of the two assets completely, which would be the only available alternative in the swap regret scenario. The conditional swap class we introduce provides precisely a way to account for such correlations between actions. We start by introducing the learning set-up and the key notions relevant to our analysis (Section 2).

## 2  Learning set-up and model

We consider the standard online learning set-up with a set of actions $\mathcal{N} = \{1, \ldots, N\}$. At each round $t \in \{1, \ldots, T\}$, $T \geq 1$, the player selects an action $x_t \in \mathcal{N}$ according to a distribution $p^t$ over $\mathcal{N}$, in response to which the adversary chooses a function $f^t \colon \mathcal{N}^t \to [0,1]$ and causes the player to incur a loss $f^t(x_t, x_{t-1}, \ldots, x_1)$. The objective of the player is to choose a sequence of actions $(x_1, \ldots, x_T)$ that minimizes his cumulative loss $\sum_{t=1}^{T} f^t(x_t, x_{t-1}, \ldots, x_1)$.

A standard metric used to measure the performance of an online algorithm $\mathcal{A}$ over $T$ rounds is its *(expected) external regret*, which measures the player's expected performance against the best fixed action in hindsight:

$$\operatorname*{Reg}_{\mathrm{Ext}}(\mathcal{A}, T) = \sum_{t=1}^{T} \mathop{\mathbb{E}}_{\substack{(x_t,..,x_1) \sim \\ (p^t,...,p^1)}} [f^t(x_t, .., x_1)] - \min_{j \in \mathcal{N}} \sum_{t=1}^{T} f^t(j, j, ..., j).$$

There are several common modifications to the above online learning scenario: (1) we may compare regret against stronger competitor classes: $\operatorname{Reg}_{\mathcal{C}}(\mathcal{A}, T) = \sum_{t=1}^{T} \mathbb{E}_{p^t,...,p^1} f^t(x_t, .., x_1) - \min_{\varphi \in \mathcal{C}} \sum_{t=1}^{T} \mathbb{E}_{p^t,...,p^1} [f^t(\varphi(x_t), \varphi(x_{t-1}), ..., \varphi(x_1))]$ for some function class $\mathcal{C} \subseteq \mathcal{N}^{\mathcal{N}}$; (2) the player may have access to only partial information about the loss, i.e. only knowledge of $f^t(x_t, .., x_1)$ as opposed to $f^t(a, x_{t-1}, \ldots, x_1) \forall a \in \mathcal{N}$ (also known as the *bandit scenario*); (3) the loss function may have bounded memory: $f^t(x_t, ..., x_{t-k}, x_{t-k-1}, ..., x_1) = f^t(x_t, ..., x_{t-k}, y_{t-k-1}, ..., y_1), \forall x_j, y_j \in \mathcal{N}$.

The scenario where $\mathcal{C} = \mathcal{N}^{\mathcal{N}}$ in (1) is called the *swap regret* case, and the case where $k = 0$ in (3) is referred to as the *oblivious adversary*. (Sublinear) regret minimization is possible for loss functions against any competitor class of the form described in (1), with only partial information, and with at least some level of bounded memory. See [4] and [1] for a reference on (1), [2] and [5] for (2), and [1] for (3). [6] also provides a detailed summary of the best known regret bounds in all of these scenarios and more.

The introduction of adversaries with bounded memory naturally leads to an interesting question: *what if we also try to increase the power of the competitor class in this way?*

While swap regret is a natural competitor class and has many useful game theoretic consequences (see [14]), one important missing ingredient is that the competitor class of functions does not have memory. In fact, in most if not all online learning scenarios and regret minimization algorithms considered so far, the point of comparison has been against modification of the player's actions at each point of time independently of the previous actions. But, as we discussed above in the financial markets example, there exist cases where a player should be measured against alternatives that depend on the past and the player should take into account the correlations between actions.

Specifically, we consider competitor functions of the form $\Phi^t \colon \mathcal{N}^t \to \mathcal{N}^t$. Let $\mathcal{C}_{\mathrm{all}} = \{\Phi^t \colon \mathcal{N}^t \to \mathcal{N}^t\}_{t=1}^{\infty}$ denote the class of all such functions. This leads us to the expression: $\sum_{t=1}^{T} \mathbb{E}_{p^1,...,p^t} [f^t] - \min_{\Phi^t \in \mathcal{C}_{\mathrm{all}}} \sum_{t=1}^{T} \mathbb{E}_{p^1,...,p^t} [f^t \circ \Phi^t]$. $\mathcal{C}_{\mathrm{all}}$ is clearly a substantially richer class of competitor functions than traditional swap regret. In fact, it is the most comprehensive class, since we can always reach $\sum_{t=1}^{T} \mathbb{E}_{p^1,...,p^t} [f^t] - \sum_{t=1}^{T} \min_{(x_1,..,x_t)} f^t(x_1, .., x_t)$ by choosing $\Phi^t$ to map all points to $\operatorname{argmin}_{(x_t,..,x_1)} f^t(x_t, ..., x_1)$. Not surprisingly, however, it is not possible to obtain a sublinear regret bound against this general class.

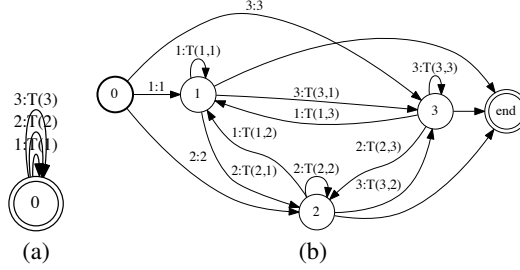

Figure 1: (a) unigram conditional swap class interpreted as a finite-state transducer. This is the same as the usual swap class and has only the trivial state; (b) bigram conditional swap class interpreted as a finite-state transducer. The action at time $t-1$ defines the current state and influences the potential swap at time $t$.

**Theorem 1.** *No algorithm can achieve sublinear regret against the class $\mathcal{C}_{\text{all}}$, regardless of the loss function's memory.*

This result is well-known in the on-line learning community, but, for completeness, we include a proof in Appendix 9. Theorem 1 suggests examining more reasonable subclasses of $\mathcal{C}_{\text{all}}$. To simplify the notation and proofs that follow in the paper, we will henceforth restrict ourselves to the scenario of an oblivious adversary, as in the original study of swap regret [4]. However, an application of the batching technique of [1] should produce analogous results in the non-oblivious case for all of the theorems that we provide.

Now consider the collection of competitor functions $\mathcal{C}_k = \{\varphi \colon \mathcal{N}^k \to \mathcal{N}\}$. Then, a player who has played actions $\{a_s\}_{s=1}^{t-1}$ in the past should have his performance compared against $\varphi(a_t, a_{t-1}, a_{t-2}, \ldots, a_{t-(k-1)})$ at time $t$, where $\varphi \in \mathcal{C}_k$. We call this class $\mathcal{C}_k$ of functions the *$k$-gram conditional swap regret class*, which also leads us to the regret definition:

$$\operatorname*{Reg}_{\mathcal{C}_k}(\mathcal{A}, T) = \sum_{t=1}^{T} \mathop{\mathbb{E}}_{x_t \sim p^t}[f^t(x_t)] - \min_{\varphi \in \mathcal{C}_k} \sum_{t=1}^{T} \mathop{\mathbb{E}}_{x_t \sim p^t}[f^t(\varphi(x_t, a_{t-1}, a_{t-2}, \ldots, a_{t-(k-1)}))].$$

Note that this is a direct extension of swap regret to the scenario where we allow for swaps conditioned on the history of the previous $(k-1)$ actions. For $k = 1$, this precisely coincides with swap regret.

One important remark about the $k$-gram conditional swap regret is that it is a random quantity that depends on the particular sequence of actions played. A natural deterministic alternative would be of the form:

$$\sum_{t=1}^{T} \mathop{\mathbb{E}}_{x_t \sim p^t}[f^t(x_t)] - \min_{\varphi \in \mathcal{C}_k} \sum_{t=1}^{T} \mathop{\mathbb{E}}_{(x_t, \ldots, x_1) \sim (p^t, \ldots, p^1)}[f^t(\varphi(x_t, x_{t-1}, x_{t-2}, \ldots, x_{t-(k-1)}))].$$

However, by taking the expectation of $\operatorname{Reg}_{\mathcal{C}_k}(\mathcal{A}, T)$ with respect to $a_{T-1}, a_{T_2}, \ldots, a_1$ and applying Jensen's inequality, we obtain

$$\operatorname*{Reg}_{\mathcal{C}_k}(\mathcal{A}, T) \geq \sum_{t=1}^{T} \mathop{\mathbb{E}}_{x_t \sim p^t}[f^t(x_t)] - \min_{\varphi \in \mathcal{C}_k} \sum_{t=1}^{T} \mathop{\mathbb{E}}_{(x_t, \ldots, x_1) \sim (p^t, \ldots, p^1)}[f^t(\varphi(x_t, x_{t-1}, x_{t-2}, \ldots, x_{t-(k-1)}))],$$

and so no generality is lost by considering the randomized sequence of actions in our regret term.

Another interpretation of the bigram conditional swap class is in the context of finite-state transducers. Taking a player's sequence of actions $(x_1, ..., x_T)$, we may view each competitor function in the conditional swap class as an application of a finite-state transducer with $N$ states, as illustrated by Figure 1. Each state encodes the history of actions $(x_{t-1}, \ldots, x_{t-(k-1)})$ and admits $N$ outgoing transitions representing the next action along with its possible modification. In this framework, the original swap regret class is simply a transducer with a single state.

## 3    Full Information Scenario

Here, we prove that it is in fact possible to minimize $k$-gram conditional swap regret against an oblivious adversary, starting with the easier to interpret bigram scenario. Our proof constructs a meta-algorithm using external regret algorithms as subroutines, as in [4]. The key is to attribute a fraction of the loss to each external regret algorithm, so that these losses sum up to our actual realized loss and also press the subroutines to minimize regret against each of the conditional swaps.

**Theorem 2.** *There exists an online algorithm $\mathcal{A}$ with bigram swap regret bounded as follows:* $\mathrm{Reg}_{\mathcal{C}_2}(\mathcal{A}, T) \leq \mathcal{O}\big(N\sqrt{T \log N}\big)$.

*Proof.* Since the distribution $p^t$ at round $t$ is finite-dimensional, we can represent it as a vector $p^t = (p_1^t, ..., p_N^t)$. Similarly, since oblivious adversaries take only $N$ arguments, we can write $f^t$ as the loss vector $f^t = (f_1^t, ..., f_N^t)$. Let $\{a_t\}_{t=1}^T$ be a sequence of random variables denoting the player's actions at each time $t$, and let $\delta_{a_t}^t$ denote the (random) Dirac delta distribution concentrated at $a_t$ and applied to variable $x_t$. Then, we can rewrite the bigram swap regret as follows:

$$
\mathrm{Reg}_{\mathcal{C}_2}(\mathcal{A}, T) = \sum_{t=1}^T \mathbb{E}_{p^t}[f^t(x_t)] - \min_{\varphi \in \mathcal{C}_2} \sum_{t=1}^T \mathbb{E}_{p^t, \delta_{a_{t-1}}^{t-1}} [f^t(\varphi(x_t, x_{t-1}))]
$$

$$
= \sum_{t=1}^T \sum_{i=1}^N p_i^t f_i^t - \min_{\varphi \in \mathcal{C}_2} \sum_{t=1}^T \sum_{i,j=1}^N p_i^t \delta_{\{a_{t-1}=j\}}^{t-1} f_{\varphi(i,j)}^t.
$$

Our algorithm for achieving sublinear regret is defined as follows:

1. At $t = 1$, initialize $N^2$ external regret minimizing algorithms $A_{i,k}$, $(i,k) \in \mathcal{N}^2$. We can view these in the form of $N$ matrices in $\mathbb{R}^{N \times N}$, $\{Q^{t,k}\}_{k=1}^N$, where for each $k \in \{1, \ldots, N\}$, $Q_i^{t,k}$ is a row vector consisting of the distribution weights generated by algorithm $A_{i,k}$ at time $t$ based on losses received at times $1, \ldots, t-1$.

2. At each time $t$, let $a_{t-1}$ denote the random action played at time $t-1$ and let $\delta_{a_{t-1}}^{t-1}$ denote the (random) Dirac delta distribution for this action. Define the $N \times N$ matrix $Q^t = \sum_{k=1}^N \delta_{\{a_{t-1}=k\}}^{t-1} Q^{t,k}$. $Q^t$ is a Markov chain (i.e., its rows sum up to one), so it admits a stationary distribution $p^t$ which we we will use as our distribution for time $t$.

3. When we draw from $p^t$, we play a random action $a_t$ and receive loss $f^t$. Attribute the portion of loss $p_i^t \delta_{\{a_{t-1}=k\}}^{t-1} f^t$ to algorithm $A_{i,k}$, and generate distributions $Q_i^{t,k}$. Notice that $\sum_{i,k=1}^N p_i^t \delta_{\{a_{t-1}=k\}}^{t-1} f^t = f^t$, so that the actual realized loss is allocated completely.

Recall that an optimal external regret minimizing algorithm $\mathcal{A}$ (e.g. randomized weighted majority) admits a regret bound of the form $R_{i,k} = R_{i,k}(L_{\min}^{i,k}, T, N) = \mathcal{O}\left(\sqrt{L_{\min}^{i,k} \log(N)}\right)$, where $L_{\min}^{i,k} = \min_{j=1}^N \sum_{t=1}^T f_j^{t,i,k}$ for the sequence of loss vectors $\{f^{t,i,k}\}_{t=1}^T$ incurred by the algorithm. Since $p^t = p^t Q^t$ is a stationary distribution, we can write:

$$
\sum_{t=1}^T p^t \cdot f^t = \sum_{t=1}^T \sum_{j=1}^N p_j^t f_j^t = \sum_{t=1}^T \sum_{j=1}^N \sum_{i=1}^N p_i^t Q_{i,j}^t f_j^t = \sum_{t=1}^T \sum_{j=1}^N \sum_{i=1}^N p_i^t \sum_{k=1}^N \delta_{\{i_{t-1}=k\}}^{t-1} Q_{i,j}^{t,k} f_j^t.
$$

Rearranging leads to

$$\sum_{t=1}^{T} p^t \cdot f^t = \sum_{i,k=1}^{N} \sum_{t=1}^{T} \sum_{j=1}^{N} p_i^t \delta_{\{i_{t-1}=k\}}^{t-1} Q_{i,j}^{t,k} f_j^t$$

$$\leq \sum_{i,k=1}^{N} \left( \left( \sum_{t=1}^{T} p_i^t \delta_{\{i_{t-1}=k\}}^{t-1} f_{\varphi(i,k)}^t \right) + R_{i,k}(L_{min},T,N) \right) \quad \text{(for arbitrary } \varphi \colon \mathcal{N}^2 \to \mathcal{N})$$

$$= \sum_{i,k=1}^{N} \left( \sum_{t=1}^{T} p_i^t \delta_{\{i_{t-1}=k\}}^{t-1} f_{\varphi(i,k)}^t \right) + \sum_{i,k=1}^{N} R_{i,k}(L_{min},T,N).$$

Since $\varphi$ is arbitrary, we obtain

$$\operatorname*{Reg}_{\mathcal{C}_2}(\mathcal{A},T) = \sum_{t=1}^{T} p^t \cdot f^t - \min_{\varphi \in \mathcal{C}_2} \sum_{t=1}^{T} \sum_{i,k=1}^{N} p_i^t \delta_{\{i_{t-1}=k\}}^{t-1} f_{\varphi(i,k)}^t \leq \sum_{i,k=1}^{N} R_{i,k}(L_{min},T,N).$$

Using the fact that $R_{i,k} = \mathcal{O}\left( \sqrt{L_{\min}^{i,k} \log(N)} \right)$ and that we scaled the losses to algorithm $A_{i,k}$ by $p_i^t \delta_{\{i_{t-1}=k\}}^{t-1}$, the following inequality holds: $\sum_{k=1}^{N} \sum_{j=1}^{N} L_{\min}^{k,j} \leq T$. By Jensen's inequality, this implies

$$\frac{1}{N^2} \sum_{k=1}^{N} \sum_{j=1}^{N} \sqrt{L_{\min}^{k,j}} \leq \sqrt{\frac{1}{N^2} \sum_{k=1}^{N} \sum_{j=1}^{N} L_{\min}^{k,j}} \leq \frac{\sqrt{T}}{N},$$

or, equivalently, $\sum_{k=1}^{N} \sum_{j=1}^{N} \sqrt{L_{\min}^{k,j}} \leq N\sqrt{T}$. Combining this with our regret bound yields

$$\operatorname*{Reg}_{\mathcal{C}_2}(\mathcal{A},T) \leq \sum_{i,k=1}^{N} R_{i,k}(L_{\min},T,N) = \sum_{i,k=1}^{N} \mathcal{O}\left( \sqrt{L_{\min}^{i,k} \log N} \right) \leq \mathcal{O}\left( N\sqrt{T \log N} \right),$$

which concludes the proof. $\qquad\square$

**Remark 1.** *The computational complexity of a standard external regret minimization algorithm such as randomized weighted majority per round is in $\mathcal{O}(N)$ (update the distribution on each of the $N$ actions multiplicatively and then renormalize), which implies that updating the $N^2$ subroutines will cost $\mathcal{O}(N^3)$ per round. Allocating losses to these subroutines and combining the distributions that they return will cost an additional $\mathcal{O}(N^3)$ time. Finding the stationary distribution of a stochastic matrix can be done via matrix inversion in $\mathcal{O}(N^3)$ time. Thus, the total computational complexity of achieving $\mathcal{O}(N\sqrt{T \log(N)})$ regret is only $\mathcal{O}(N^3 T)$. We remark that in practice, one often uses iterative methods to compute dominant eigenvalues (see [16] for a standard reference and [11] for recent improvements). [10] has also studied techniques to avoid computing the exact stationary distribution at every iteration step for similar types of problems.*

The meta-algorithm above can be interpreted in three equivalent ways: (1) the player draws an action $x_t$ from distribution $p^t$ at time $t$; (2) the player uses distribution $p^t$ to choose among the $N$ subsets of algorithms $Q_1^t, ..., Q_N^t$, picking one subset $Q_j^t$; next, after drawing $j$ from $p^t$, the player uses $\delta_{\{a_{t-1}=k\}}^{t-1}$ to randomly choose among the algorithms $Q_j^{t,1}, ..., Q_j^{t,N}$, picking algorithm $Q_j^{t,a_{t-1}}$; after locating this algorithm, the player uses the distribution from algorithm $Q_j^{t,a_{t-1}}$ to draw an action; (3) the player chooses algorithm $Q_j^{t,k}$ with probability $p_j^t \delta_{\{a_{t-1}=k\}}^{t-1}$ and draws an action from its distribution.

The following more general bound can be given for an arbitrary $k$-gram swap scenario.

**Theorem 3.** *There exists an online algorithm $\mathcal{A}$ with $k$-gram swap regret bounded as follows:* $\operatorname*{Reg}_{\mathcal{C}_k}(\mathcal{A},T) \leq \mathcal{O}\left( \sqrt{N^k T \log N} \right).$

The algorithm used to derive this result is a straightforward extension of the algorithm provided in the bigram scenario, and the proof is given in Appendix 11.

**Remark 2.** *The computational complexity of achieving the above regret bound is $O(N^{k+1}T)$.*

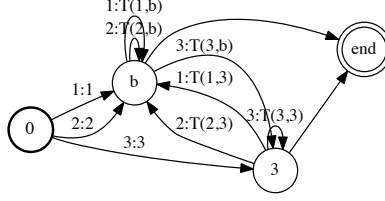

Figure 2: bigram conditional swap class restricted to a finite number of active states. When the action at time $t-1$ is 1 or 2, the transducer is in the same state, and the swap function is the same.

## 4 State-Dependent Bounds

In some situations, it may not be relevant to consider conditional swaps for every possible action, either because of the specific problem at hand or simply for the sake of computational efficiency. Thus, for any $\mathcal{S} \subseteq \mathcal{N}^2$, we define the following competitor class of functions:

$$\mathcal{C}_{2,\mathcal{S}} = \{\varphi \colon \mathcal{N}^2 \to \mathcal{N} | \varphi(i,k) = \tilde{\varphi}(i) \text{ for } (i,k) \in \mathcal{S} \text{ where } \tilde{\varphi} \colon \mathcal{N} \to \mathcal{N}\}.$$

See Figure 2 for a transducer interpretation of this scenario.

We will now show that the algorithm above can be easily modified to derive a tighter bound that is dependent on the number of states in our competitor class. We will focus on the bigram case, although a similar result can be shown for the general $k$-gram conditional swap regret.

**Theorem 4.** *There exists an online algorithm $\mathcal{A}$ such that* $\operatorname{Reg}_{\mathcal{C}_{2,\mathcal{S}}}(\mathcal{A}, T) \leq \mathcal{O}(\sqrt{T(|\mathcal{S}^c| + N)\log(N)})$.

The proof of this result is given in Appendix 10. Note that when $S = \emptyset$, we are in the scenario where all the previous states matter, and our bound coincides with that of the previous section.

**Remark 3.** *The computational complexity of achieving the above regret bound is $O((N(|\pi_1(\mathcal{S})| + |\mathcal{S}^c|) + N^3)T)$, where $\pi_1$ is projection onto the first component. This follows from the fact that we allocate the same loss to all $\{A_{i,k}\}_{k:(i,k)\in\mathcal{S}} \; \forall i \in \pi_1(\mathcal{S})$, so we effectively only have to manage $|\pi_1(\mathcal{S})| + |\mathcal{S}^c|$ subroutines.*

## 5 Conditional Correlated Equilibrium and $\epsilon$-Dominated Actions

It is well-known that regret minimization in on-line learning is related to game-theoretic equilibria [14]. Specifically, when both players in a two-player zero-sum game follow external regret minimizing strategies, then the product of their individual empirical distributions converges to a Nash equilibrium. Moreover, if all players in a general $K$-player game follow swap regret minimizing strategies, then their empirical joint distribution converges to a correlated equilibrium [7].

We will show in this section that when all players follow conditional swap regret minimization strategies, then the empirical joint distribution will converge to a new stricter type of correlated equilibrium.

**Definition 1.** *Let $\mathcal{N}_k = \{1, ..., N_k\}$, for $k \in \{1, ..., K\}$ and $G = (\mathcal{S} = \times_{k=1}^{K} \mathcal{N}_k, \{l^{(k)} \colon \mathcal{S} \to [0,1]\}_{k=1}^{K})$ denote a $K$-player game. Let $s = (s_1, ..., s_K) \in \mathcal{S}$ denote the strategies of all players in one instance of the game, and let $s_{(-k)}$ denote the $(K-1)$-vector of strategies played by all players aside from player $k$. A joint distribution $P$ on two rounds of this game is a **conditional correlated equilibrium** if for any player $k$, actions $j, j' \in \mathcal{N}_k$, and map $\varphi_k \colon \mathcal{N}_k^2 \to \mathcal{N}_k$, we have*

$$\sum_{(s,r)\in\mathcal{S}^2 \colon s_k=j, r_k=j'} P(s,r) \left(l^{(k)}(s_k, s_{(-k)}) - l^{(k)}(\varphi_k(s_k, r_k), s_{(-k)})\right) \leq 0.$$

The standard interpretation of correlated equilibrium, which was first introduced by Aumann, is a scenario where an external authority assigns mixed strategies to each player in such a way that no player has an incentive to deviate from the recommendation, provided that no other player deviates

from his [3]. In the context of repeated games, a conditional correlated equilibrium is a situation where an external authority assigns mixed strategies to each player in such a way that no player has an incentive to deviate from the recommendation in the second round, even after factoring in information from the previous round of the game, provided that no other player deviates from his.

It is important to note that the concept of conditional correlated equilibrium presented here is different from the notions of extensive form correlated equilibrium and repeated game correlated equilibrium that have been studied in the game theory and economics literature [8, 12].

Notice that when the values taken for $\varphi_k$ are indepndent of its second argument, we retrieve the familiar notion of correlated equilibrium.

**Theorem 5.** *Suppose that all players in a $K$-player repeated game follow bigram conditional swap regret minimizing strategies. Then, the joint empirical distribution of all players converges to a conditional correlated equilibrium.*

*Proof.* Let $I^t \in \mathcal{S}$ be a random vector denoting the actions played by all $K$ players in the game at round $t$. The empirical joint distribution of every two subsequent rounds of a $K$-player game played repeatedly for $T$ total rounds has the form $\widehat{P}^T = \frac{1}{T}\sum_{t=1}^{T}\sum_{(s,r)\in\mathcal{S}^2}\delta_{\{I^t=s, I^{t-1}=r\}}$, where $I = (I_1, .., I_K)$ and $I_k \sim p^{(k)}$ denotes the action played by player $k$ using the mixed strategy $p^{(k)}$.

Let $q^{t,(k)}$ denote $\delta_{\{i_{t-1}=k\}}^{t-1} \otimes p^{t-1,(k-1)}$. Then, the conditional swap regret of each player $k$, $\text{reg}(k,T)$, can be bounded as follows since he is playing with a conditional swap regret minimizing strategy:

$$\text{reg}(k,T) = \frac{1}{T}\sum_{t=1}^{T}\mathbb{E}_{s_k^t \sim p^{t,(k)}}\left[l^{(k)}(s_k, s_{(-k)})\right] - \min_{\varphi}\frac{1}{T}\sum_{t=1}^{T}\mathbb{E}_{\substack{(s_k^t, s_k^{t-1})\\ \sim\left(p^{t,(k)}, q^{t,(k)}\right)}}\left[l^{(k)}(\varphi(s_k^t, s_k^{t-1}), s_{(-k)}^t)\right]$$

$$\leq \mathcal{O}\left(N\sqrt{\frac{\log(N)}{T}}\right).$$

Define the instantaneous conditional swap regret vector as

$$\widehat{r}_{t,j_0,j_1}^{(k)} = \delta_{\{I_{(k)}^t=j_0, I_{(k)}^{t-1}=j_1\}}\left(l^{(k)}(I^t) - l^{(k)}(\varphi_k(j_0, j_1), I_{(-k)}^t)\right),$$

and the expected instantaneous conditional swap regret vector as

$$r_{t,j_0,j_1}^{(k)} = \mathbb{P}(s_k^t = j_0)\delta_{\{I_{(k)}^{t-1}=j_1\}}\left(l^{(k)}(j_0, I_{(-k)}^t) - l^{(k)}(\varphi_k(j_0, j_1), I_{(-k)}^t)\right).$$

Consider the filtration $G_t = \{$information of opponents at time $t$ and of the player's actions up to time $t-1\}$. Then, we see that $\mathbb{E}\left[\widehat{r}_{t,j_0,j_1}^{(k)}|G_t\right] = r_{t,j_0,j_1}^{(k)}$. Thus, $\{R_t = r_{t,j_0,j_1}^{(k)} - \widehat{r}_{t,j_0,j_1}^{(k)}\}_{t=1}^{\infty}$ is a sequence of bounded martingale differences, and by the Hoeffding-Azuma inequality, we can write for any $\alpha > 0$, that $\mathbb{P}[|\sum_{t=1}^{T}R_t| > \alpha] \leq 2\exp(-C\alpha^2/T)$ for some constant $C > 0$.

Now define the sets $A_T := \left\{\left|\frac{1}{T}\sum_{t=1}^{T}R_t\right| > \sqrt{\frac{C}{T}\log\left(\frac{2}{\delta_T}\right)}\right\}$. By our concentration bound, we have $P(A_T) \leq \delta_T$. Setting $\delta_T = \exp(-\sqrt{T})$ and applying the Borel-Cantelli lemma, we obtain that $\limsup_{T\to\infty}|\frac{1}{T}\sum_{t=1}^{T}R_t| = 0$ a.s..

Finally, since each player followed a conditional swap regret minimizing strategy, we can write $\limsup_{T\to\infty}\frac{1}{T}\sum_{t=1}^{T}\widehat{r}_{t,j_0,j_1}^{(k)} \leq 0$. Now, if the empirical distribution did not converge to a conditional correlated equilibrium, then by Prokhorov's theorem, there exists a subsequence $\{\widehat{P}^{T_j}\}_j$ satisfying the conditional correlated equilibrium inequality but converging to some limit $P^*$ that is not a conditional correlated equilibrium. This cannot be true because the inequality is closed under weak limits. $\square$

Convergence to equilibria over the course of repeated game-playing also naturally implies the scarcity of "very suboptimal" strategies.

**Definition 2.** *An action pair $(s_k, r_k) \in \mathcal{N}_k^2$ played by player $k$ is* **conditionally $\epsilon$-dominated** *if there exists a map $\varphi_k \colon \mathcal{N}_k^2 \to \mathcal{N}_k$ such that*

$$l^{(k)}(s_k, s_{(-k)}) - l^{(k)}(\varphi_k(s_k, r_k), s_{(-k)}) \geq \epsilon.$$

**Theorem 6.** *Suppose player $k$ follows a conditional swap regret minimizing strategy that produces a regret $R$ over $T$ instances of the repeated game. Then, on average, an action pair of player $k$ is conditionally $\epsilon$-dominated at most $\frac{R}{\epsilon T}$ fraction of the time.*

The proof of this result is provided in Appendix 12.

## 6 Bandit Scenario

As discussed earlier, the bandit scenario differs from the full-information scenario in that the player only receives information about the loss of his action $f^t(x_t)$ at each time and not the entire loss function $f^t$. One standard external regret minimizing algorithm is the Exp3 algorithm introduced by [2], and it is the base learner off of which we will build a conditional swap regret minimizing algorithm.

To derive a sublinear conditional swap regret bound, we require an external regret bound on Exp3:

$$\sum_{t=1}^{T} \mathbb{E}_{p_t}[f^t(x_t)] - \min_{a \in \mathcal{N}} \sum_{t=1}^{T} f^t(a) \leq 2\sqrt{L_{\min} N \log(N)},$$

which can be found in Theorem 3.1 of [5]. Using this estimate, we can derive the following result.

**Theorem 7.** *There exists an algorithm $\mathcal{A}$ such that $\mathrm{Reg}_{\mathcal{C}_2, bandit}(\mathcal{A}, T) \leq \mathcal{O}\big(\sqrt{N^3 \log(N) T}\big)$.*

The proof is given in Appendix 13 and is very similar to the proof for the full information setting.

It can also easily be extended in the analogous way to provide a regret bound for the $k$-gram regret in the bandit scenario.

**Theorem 8.** *There exists an algorithm $\mathcal{A}$ such that $\mathrm{Reg}_{\mathcal{C}_k, bandit}(\mathcal{A}, T) \leq \mathcal{O}\big(\sqrt{N^{k+1} \log(N) T}\big)$.*

See Appendix 14 for an outline of the algorithm.

## 7 Conclusion

We analyzed the extent to which on-line learning scenarios are learnable. In contrast to some of the more recent work that has focused on increasing the power of the adversary (see e.g. [1]), we increased the power of the competitor class instead by allowing history-dependent action swaps and thereby extending the notion of swap regret. We proved that this stronger class of competitors can still be beaten in the sense of sublinear regret as long as the memory of the competitor is bounded. We also provided a state-dependent bound that gives a more favorable guarantee when only some parts of the history are considered. In the bigram setting, we introduced the notion of conditional correlated equilibrium in the context of repeated $K$-player games, and showed how it can be seen as a generalization of the traditional correlated equilibrium. We proved that if all players follow bigram conditional swap regret minimizing strategies, then the empirical joint distribution converges to a conditional correlated equilibrium and that no player can play very suboptimal strategies too often. Finally, we showed that sublinear conditional swap regret can also be achieved in the partial information bandit setting.

## 8 Acknowledgements

We thank the reviewers for their comments, many of which were very insightful. We are particularly grateful to the reviewer who found an issue in our discussion on conditional correlated equilibrium and proposed a helpful resolution. This work was partly funded by the NSF award IIS-1117591. The material is also based upon work supported by the National Science Foundation Graduate Research Fellowship under Grant No. DGE 1342536.

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
