[Supplementary Material]

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

*Proof.* If sublinear regret is impossible in the oblivious case, then it is impossible for any level of memory. Now, for any $t$, let $j_t^* \in \arg\min_j p_j^t$ and define for the adversary the following sequence of loss functions:

$$f^t(x^t) = \begin{cases} 1 & \text{for } x^t \neq j_t^* \\ 0 & \text{for } x^t = j_t^*. \end{cases}$$

Then, the following holds:

$$\sum_{t=1}^{T} \mathbb{E}_{p^1,\dots,p^t}[f^t] - \sum_{t=1}^{T} \min_{(x_1,\dots,x_t)} f^t(x_1,\dots,x_t)$$

$$= \sum_{t=1}^{T} \mathbb{E}_{p^t}[f^t] - \sum_{t=1}^{T} \min_{x_t} f^t(x_t)$$

$$= \sum_{t=1}^{T} \mathbb{E}_{p^t}[f^t] = \sum_{t=1}^{T} 1 - \min_j p_j \geq \sum_{t=1}^{T} 1 - 1/N = T(1 - 1/N) = \Omega(T),$$

which concludes the proof. $\qquad\square$

## 10 Proof of Theorem 4

**Theorem 4.** *There exists an online algorithm $\mathcal{A}$ such that $\text{Reg}_{\mathcal{C}_{2,\mathcal{S}}}(\mathcal{A}, T) \leq \mathcal{O}(\sqrt{T(|\mathcal{S}^c| + N)\log(N)})$.*

*Proof.* The cumulative loss can be written as follows in terms of $\mathcal{S}$:

$$\sum_{t=1}^{T} p^t \cdot f^t = \sum_{i,k=1}^{N} \sum_{t=1}^{T} \sum_{j=1}^{N} p_i^t \delta_{\{a_{t-1}=k\}}^{t-1} Q_{i,j}^{t,k} f_j^t = \sum_{t=1}^{T} \left( \sum_{(i,k)\in\mathcal{S}} + \sum_{(i,k)\notin\mathcal{S}} \right) \sum_{j=1}^{N} p_i^t \delta_{\{a_{t-1}=k\}}^{t-1} Q_{i,j}^{t,k} f_j^t.$$

Our algorithm $\mathcal{A}$ is the same as the one in the bigram case with the one caveat that all subroutines $A_{i,k}$ must be derived from the same external regret minimizing algorithm. Then, as in the previous section, we can derive the bound

$$\sum_{t=1}^{T} \sum_{(i,k)\notin\mathcal{S}} \sum_{j=1}^{N} p_i^t \delta_{\{a_{t-1}=k\}}^{t-1} Q_{i,j}^{t,k} f_j^t \leq \sum_{(i,k)\notin\mathcal{S}} \left( \sum_{t=1}^{T} p_i^t \delta_{\{a_{t-1}=k\}}^{t-1} f_{\varphi(i,k)}^t + R_{i,k}(L_{\min}, T, N) \right).$$

For the action pairs $(i,k) \in \mathcal{S}$, we can impose $Q_{i,j}^{t,k} = Q_{i,j}^{t,0}$ for all $k$, by associating to all algorithms $A_{i,k}$ the same loss $\sum_{k:\,(i,k)\in\mathcal{S}} f_j^t p_i^t \delta_{\{a_{t-1}=k\}}^{t-1}$ and using the fact that all $A_{i,k}$ are based off of the same subroutine.

With this choice of loss allocation, we can write

$$\sum_{t=1}^{T} \sum_{(i,k)\in\mathcal{S}} \sum_{j=1}^{N} p_i^t \delta_{\{a_{t-1}=k\}}^{t-1} Q_{i,j}^{t,k} f_j^t$$

$$= \sum_{t=1}^{T} \sum_{i:\,\exists k,(i,k)\in\mathcal{S}} \sum_{j=1}^{N} \left( \sum_{k:\,(i,k)\in\mathcal{S}} p_i^t \delta_{\{a_{t-1}=k\}}^{t-1} f_j^t \right) Q_{i,j}^{t,0}$$

$$\leq \sum_{i:\,\exists k,(i,k)\in\mathcal{S}} \left( \sum_{t=1}^{T} \left( \sum_{k:\,(i,k)\in\mathcal{S}} p_i^t \delta_{\{a_{t-1}=k\}}^{t-1} f_{\tilde{\varphi}(i)}^t \right) + R_i(L_{\min}, T, N) \right).$$

Combining the two terms yields

$$\sum_{t=1}^{T} p^t \cdot f^t \leq \sum_{t=1}^{T} \sum_{i,k=1}^{N} p_i^t \delta_{\{a_{t-1}=k\}}^{t-1} f_{\varphi(i,k)}^t + \sum_{(i,k)\notin\mathcal{S}} R_{i,k}(L_{\min}, T, N) + \sum_{i:\ \exists k,(i,k)\in\mathcal{S}} R_i(L_{\min,T,N}).$$

Next, using the fact that we allocated the original loss over the sub-algorithms, we can apply the standard bounds on external regret algorithms to obtain $\mathrm{Reg}_{\mathcal{C}_{2,\mathcal{S}},\mathrm{Obliv}}(\mathcal{A}, T) \leq \mathcal{O}(\sqrt{T(|\mathcal{S}^c| + N)\log(N)})$. $\qquad\square$

## 11   Proof of Theorem 3

**Theorem 3.** *There exists an online algorithm $\mathcal{A}$ with $k$-gram swap regret bounded as follows:* $\mathrm{Reg}_{\mathcal{C}_k}(\mathcal{A}, T) \leq \mathcal{O}(\sqrt{N^k T \log N})$.

*Proof.* The result follows from a natural extension of the algorithm used in Theorem 2.

1. At $t = 1$, initialize $N^k$ external regret minimizing algorithms indexed as $\{A_{j_0,..,j_{k-1}}\}_{j_0,..,j_{k-1}=1}^{N}$. This defines $N^{k-1}$ matrices in $\mathbb{R}^{N\times N}$, $\{Q^{t,j_1,...,j_{k-1}}\}_{j_1,...,j_{k-1}=1}^{N}$, where, for each fixed $j_0,\ldots,j_{k-1}$, $Q_{j_0}^{t,j_1,...,j_{k-1}}$ is a row vector corresponding to the distribution generated by algorithm $A_{j_0,..,j_{k-1}}$ at time $t$ based on the losses it received at times $1,\ldots,t-1$.

2. At each time $t$, let $\{a_s\}_{s=1}^{t-1}$ denote the sequence of random actions played at times $1, 2, \ldots, t-1$ and let $\{\delta_{a_s}^s\}_{s=1}^{t-1}$ denote a sequence of (random) Dirac delta distributions corresponding to these actions. Define the $N \times N$ matrix

$$Q^t = \sum_{j_1,j_2,\ldots,j_{k-1}=1}^{N} \delta_{\{a_{t-1}=j_1\}}^{t-1} \delta_{\{a_{t-2}=j_2\}}^{t-2} \cdots \delta_{\{a_{t-(k-1)}=j_{k-1}\}}^{t-(k-1)} Q^{t,j_1,...,j_{k-1}}.$$

$Q^t$ is a Markov chain (i.e. its rows sum up to one), so it admits a stationary distribution $p^t$ which we we will use as our distribution for time $t$.

3. When we draw from $p^t$, we play a random action $a_t$ and receive loss $f^t$. Attribute the portion of loss $\left(p_{j_0}^t \delta_{\{a_{t-1}=j_1\}}^{t-1} \cdots \delta_{\{a_{t-(k-1)}=j_{k-1}\}}^{t-(k-1)} f^t\right)$ loss to algorithm $A_{j_0,..,j_{k-1}}$, and generate distributions $Q_{j_0}^{t,j_1,...,j_{k-1}}$.

Using this distribution and proceeding otherwise as in the proof of Theorem 2 to bound the cumulative loss leads to the desired inequality.

$\qquad\square$

## 12   Proof of Theorem 6

**Theorem 6.** *Suppose player $k$ follows a conditional swap regret minimizing strategy that produces a regret $R$ over $T$ instances of the repeated game. Then, on average, an action pair of player $k$ is conditionally $\epsilon$-dominated at most $\frac{R}{\epsilon T}$ fraction of the time.*

*Proof.* Let

$$D_\epsilon = \{(s,r) \in \mathcal{N}_k^2 \mid \exists \varphi_k \colon \mathcal{N}_k^2 \to \mathcal{N}_k \text{ s.t. } l^{(k)}(s_k, s_{(-k)}) - l^{(k)}(\varphi_k(s_k, r_k), s_{(-k)}) \geq \epsilon\}$$

denote the set of action pairs that are conditionally $\epsilon$-dominated. Then $\widehat{P}^T(D_\epsilon) = \frac{1}{T}\sum_{t=1}^{T}\sum_{(s,r)\in D_\epsilon}\delta_{\{s_k^t=s,s_k^{t-1}=r\}}$ is the total empirical mass of $D_\epsilon$, and we have

$$
\begin{aligned}
\epsilon T \widehat{P}^T(D_\epsilon) &= \epsilon \sum_{t=1}^{T}\sum_{(s,r)\in D_\epsilon}\delta_{\{s_k^t=s,s_k^{t-1}=r\}}\\
&\leq \sum_{t=1}^{T}\sum_{(s,r)\in D_\epsilon}\mathbb{E}[l^{(k)}(s_k^t,s_{(-k)}^t) - l^{(k)}(\varphi_k(s_k^t,s_k^{t-1}),s_{(-k)}^t)]\\
&\leq \max_{\varphi}\sum_{t=1}^{T}\sum_{(s,r)\in D_\epsilon}\mathbb{E}[l^{(k)}(s_k^t,s_{(-k)}^t) - l^{(k)}(\varphi(s_k^t,s_k^{t-1}),s_{(-k)}^t)]\\
&= \operatorname{reg}(k,T),
\end{aligned}
$$

and the conditional swap regret of player $k$, $\operatorname{reg}(k,T)$, satisfies $\operatorname{reg}(k,T) \geq \epsilon T \widehat{P}^T(D_\epsilon)$. Furthermore, since player $k$ is following a conditional swap regret minimizing strategy, we must have $\operatorname{reg}(k,T) \leq R$. This implies that $R \geq \epsilon T \widehat{P}^T(D_\epsilon)$. $\qquad\square$

## 13  Proof of Theorem 7

**Theorem 7.** *There exists an algorithm $\mathcal{A}$ such that $\operatorname{Reg}_{\mathcal{C}_2,bandit}(\mathcal{A},T) \leq \mathcal{O}\big(\sqrt{N^3\log(N)T}\big)$.*

*Proof.* As in the full information scenario, we will construct our distribution $p^t$ as follows:

1. At $t = 1$, initialize $N^2$ external regret minimizing algorithms $A_{i,k}$, $(i,k) \in \mathcal{N}^2$. We can view these in the form of $N$ matrices in $\mathbb{R}^{N\times N}$, $\{Q^{t,k}\}_{k=1}^{N}$, where for each $k \in \{1,\dots,N\}$, $Q_i^{t,k}$ is a row vector consisting of the distribution generated by algorithm $A_{i,k}$ at time $t$ based on losses received at times $1,\dots,t-1$.

2. At each time $t$, let $a_{t-1}$ denote the random action played at time $t-1$ and let $\delta_{a_{t-1}}^{t-1}$ denote the (random) Dirac delta distribution for this action. Define the $N \times N$ matrix $Q^t = \sum_{k=1}^{N}\delta_{\{a_{t-1}=k\}}^{t-1}Q^{t,k}$. $Q^t$ is a Markov chain (i.e., its rows sum up to one), so it admits a stationary distribution $p^t$ which we we will use as our distribution for time $t$.

3. When we draw from $p^t$, we play a random action $a_t$ and receive loss $f_{a_t}^t$. Attribute the portion of loss $p_i^t\delta_{\{a_{t-1}=k\}}^{t-1}f_{a_t}^t$ loss to algorithm $A_{i,k}$, and generate distributions $Q_i^{t,k}$ from the algorithms.

This algorithm allows us to compute

$$
\begin{aligned}
\sum_{t=1}^{T}\mathbb{E}_{x_t\sim p^t}[l_{x_t}^t] &= \sum_{t=1}^{T}\sum_{i=1}^{N}p_i^t l_i^t = \sum_{t=1}^{T}\sum_{i,k,l=1}^{N}p_k^t\delta_{\{a_{t-1}=l\}}^{t-1}Q_{k,i}^{t,l}l_i^t\\
&= \sum_{k,l=1}^{N}\sum_{t=1}^{T}\sum_{i=1}^{N}\left(p_k^t\delta_{\{a_{t-1}=l\}}^{t-1}l_i^t\right)Q_{k,i}^{t,l} = \sum_{k,l=1}^{N}\sum_{t=1}^{T}\mathbb{E}_{a_t^{k,l}\sim Q_k^{t,l}}\left[p_k^t\delta_{\{a_{t-1}=l\}}^{t-1}l_{a_t^{k,l}}^t\right]\\
&\leq \sum_{k,l=1}^{N}\sum_{t=1}^{T}\left[\left(\sum_{t=1}^{T}p_k^t\delta_{\{a_{t-1}=l\}}^{t-1}l_{\varphi(k,l)}^t\right) + 2\sqrt{L_{\min}^{k,l}N\log(N)}\right]\\
&= \sum_{t=1}^{T}\mathbb{E}_{(j_t,j_{t-1})\sim(p^t,\delta_{i_{t-1}}^{t-1})}\left[l_{\varphi(j_t,j_{t-1})}^t\right] + 2N^2\sqrt{L_{\min}^{k,l}N\log(N)},
\end{aligned}
$$

where the inequality comes from estimate (3.3) provided by Theorem 3.1 in [5]. By applying the same convexity argument as in Theorem 2, we can refine the bound to get

$$\sum_{t=1}^{T} \mathbb{E}_{p^t}[l_{i_t}^t] - \sum_{t=1}^{T} \mathbb{E}_{(j_t, j_{t-1}) \sim (p^t, \delta_{i_{t-1}}^{t-1})}[l_{\varphi(j_t, j_{t-1})}^t] \leq 2\sqrt{N^2 L_{\min} N \log(N)}$$

$$\leq 2\sqrt{N^3 T \log(N)}$$

as desired.

$\square$

## 14 Proof of Theorem 8

**Theorem 8.** *There exists an algorithm $\mathcal{A}$ such that* $\text{Reg}_{\mathcal{C}_2, bandit}(\mathcal{A}, T) \leq \mathcal{O}\left(\sqrt{N^{k+1} \log(N) T}\right)$.

*Proof.* As in the full information case, the result follows from a natural extension of the algorithm used in Theorem 7 and is analogous to the algorithm used in THeorem 3.

1. At $t = 1$, initialize $N^k$ external regret minimizing algorithms indexed as $\{A_{j_0,..,j_{k-1}}\}_{j_0,..,j_{k-1}=1}^{N}$. This defines $N^{k-1}$ matrices in $\mathbb{R}^{N \times N}$, $\{Q^{t,j_1,...,j_{k-1}}\}_{j_1,...,j_{k-1}=1}^{N}$, where for each fixed $j_0, \ldots, j_{k-1}$, $Q_{j_0}^{t,j_1,...,j_{k-1}}$ is a row vector corresponding to the distribution generated by algorithm $A_{j_0,..,j_{k-1}}$ at time $t$ based on the losses it received at times $1, \ldots, t-1$.

2. At each time $t$, let $\{a_s\}_{s=1}^{t-1}$ denote the sequence of random actions played at times $1, 2, \ldots, t-1$ and let $\{\delta_{a_s}^s\}_{s=1}^{t-1}$ denote a sequence of (random) Dirac delta distributions corresponding to these actions. Define the $N \times N$ matrix

$$Q^t = \sum_{j_1, j_2, \ldots, j_{k-1}=1}^{N} \delta_{\{a_{t-1}=j_1\}}^{t-1} \delta_{\{a_{t-2}=j_2\}}^{t-2} \cdots \delta_{\{a_{t-(k-1)}=j_{k-1}\}}^{t-(k-1)} Q^{t,j_1,...,j_{k-1}}.$$

   $Q^t$ is a Markov chain (i.e. its rows sum up to one), so it admits a stationary distribution $p^t$ which we we will use as our distribution for time $t$.

3. When we draw from $p^t$, we play a random action $a_t$ and receive loss $f_{a_t}^t$. Attribute the portion of loss $\left(p_{j_0}^t \delta_{\{a_{t-1}=j_1\}}^{t-1} \cdots \delta_{\{a_{t-(k-1)}=j_{k-1}\}}^{t-(k-1)} f_{a_t}^t\right)$ loss to algorithm $A_{j_0,..,j_{k-1}}$, and generate distributions $Q_{j_0}^{t,j_1,...,j_{k-1}}$.

Using this distribution and proceeding otherwise as in the proof of Theorem 7 to bound the cumulative loss leads to the desired inequality. $\square$