[Reviews · NeurIPS 2014]

Submitted by Assigned_Reviewer_9

The paper investigates the problem of online learning for minimizing a
generalized version of swap regret. More precisely, the authors consider
the notion of conditional swap regret, where the swap regret is defined for
a stronger adversary than usual in the sense that the adversary's action
depends on the past sequence of the player.
In particular, when the memory size of the adversary is restricted to k,
the regret is called the k-gram conditional regret. The authors propose
prediction strategies with a k-gram conditional regret of O(\sqrt{N^k T log N})
and state-dependent regret bound, respectively. Moreover, using the conditional
swap regret, the authors defines the conditional correlated equilibrium and
shows a convergence result. Finally, the authors also propose a bandit
algorithm with a 2-gram conditional swap regret bound.

The paper is well organized and well written.
The main technique for deriving the k-gram conditional swap regret is a
reduction to the analysis of N^k runs of online prediction algorithms for
oblivious adversary. The technical result is solid, although the proof
technique looks rather standard.

My concern is the time complexity of the proposed algorithms.
Apparently, the time depends on O(N^k), which looks too expensive.
The question is whether it is possible to reduce the complexity for some
advantageous cases. Also, I wonder if the time complexity O(|S_C|) is
achieved in the state-dependent case.

It would be beneficial for readers if the authors discuss the relationship
between this result and previous results on adaptive adversary.
Summary: The problem is well defined and the technical result is solid.

Submitted by Assigned_Reviewer_15

This paper discusses a type of swap regret which is conditioned on past actions of the user. They connected this concept with a new type of correlated equilibria. They designed hardness results for having the behavior conditional on the whole past, and came up with an algorithm which minimized the new type of regret. Finally, it showed how to deal with this type of regret in a bandit setting.

First of all, I think that this is an interesting area of research. However, I didn't see what I look for in new concepts of equilibrium, namely whether one can cooperate with one's opponent and still minimize regret. Secondly, I found that the paper was unclear when it could have been done so much more cleanly. Specifically, instead of a convoluted description of interleaving multiple regret algorithms, they could have simply started from the algorithms based upon regret matching and swap regret.

Specifically, you can break down the regret into "regret on time T from playing action i after action j instead of action k".

r^T_{i,j,k}=p^T_i p^{T-1}_j (f^T_i - f^T_k)

Then, define the total regret as:

R^T_{i,j,k}=\sum_{t=2}^T r^t_{i,j,k}

Then, bigram swap regret is:

Regret_T=\sum_{i} \max_{k} R^T_{i,j,k}

Thus, if all of the R^T_{i,j,k} are increasing sublinearly, you're good. Define:

R^{T,+}_{i,j,k}=max(0, R^T_{i,j,k})

Blackwell's approachability theorem needs:

\sum_{i,j,k} r^{T+1}_{i,j,k} R^{T,+}_{i,j,k} \leq 0

which implies:

\sum_{i,j,k} p^{T+1}_i p^{T}_j (f^{T+1}_i - f^{T+1}_k) R^{T,+}_{i,j,k} \leq 0

We know p^T and R^{T,+}, so we pull that to the right:

\sum_{i,k} p^{T+1}_i (f^{T+1}_i - f^{T+1}_k) \sum_j p^{T}_j R^{T,+}_{i,j,k} \leq 0

You can define a value:

R^{T,+}_{i,k} = \sum_j p^{T}_j R^{T,+}_{i,j,k}

So, our new desired inequality is:
\sum_{i,k} p^{T+1}_i (f^{T+1}_i - f^{T+1}_k) R^{T,+}_{i,k} \leq 0
This is exactly swap regret. Moreover, it extends to n-grams easily (r^T_{i,j,k,l}=p^T_i p^{T_1}_j p^{T-2}_k (f_l - f_i) for trigams). If you don't care about a particular transition in a particular state (as in state-dependent bounds), you can zero out that R^T_{i,j,k}.

T_{i,j,k}. So, this seems like a much simpler way to do what they are trying to do.

Summary: I thought this paper had good ideas. I found them unnecessarily confusing, avoiding easy standard notation. Although they used a more novel way of thinking about regret, by focusing on oblivious algorithms, they limited the impact of the paper. I don't have a problem accepting this paper, but it didn't thrill me either.

Submitted by Assigned_Reviewer_38

The paper introduces a new notion of regret, called conditional swap regret, that generalizes the notion of swap regret to action modifications conditioned on history. The authors prove regret bounds that show that learning is possible in this more general setting. The regret grows exponentially with the length of history. A regret bound is also shown for the bandit setting. Further, connections with a new notion of equilibrium, conditional correlated equilibrium, are shown: if all players follow conditional swap regret minimization methods, then their distributions converge to the conditional correlated equilibrium.

These are interesting theoretical results and the proofs seem correct.

Section 2 talks about history-dependent loss functions in detail, although only memory-less losses are considered in this work. This causes some confusion as the competitor class is history-dependent (but not losses). I suggest that authors rewrite this section and omit the policy regret part.
Summary: The paper studies an interesting problem and makes a nice contribution.
Author Feedback
Author rebuttal: Response to Reviewer 15:
Thank you for your thoughtful remarks.

We understand why you may not like the equilibrium that we introduce, but we hope you notice that since swap regret leads to correlated equilibrium, and the latter does not presuppose any cooperation among players (aside from having them adhere to some correlation device), it is natural that conditional swap regret would lead to a similar type of equilibrium as well.

It is unfortunate that you found some of the concepts in the paper unclear. With respect to notation, if you look at some of the references that we cite, (e.g. [1], [2], [4]), you will see that we largely followed the notation provided in those papers.

With respect to our proofs, from our understanding, you would have preferred us to define an instantaneous regret term corresponding to conditional swap regret, build up a condition for Blackwell approachability, and then show that we can find a requisite distribution. That sounds like an interesting alternative approach, and we will mention it in the final version of the paper. However, our methods build off results and ideas from the original paper introducing swap regret in [4], so we view them to be very natural. While you are correct that there are multiple regret algorithms being incorporated, we take great care to illustrate the main ideas in lines 235-244.

To respond to your last comment about oblivious adversaries, while it is true that this limits the power of the learner, we point out in lines 150-156 a scenario in which conditional swap regret with oblivious adversaries actually makes more sense than with adaptive adversaries. Moreover, we mention in the conclusion that we plan to study the scenario with adaptive adversaries in future work.

To Reviewer 38
Thank you for your helpful suggestions.

In section 2, we included the discussion about history-dependent loss functions to illustrate how much of the recent work has been on focused on increasing the power of the loss function, while our work, in contrast, looked at increasing the power of the competitor class. This was a point we recapitulated in lines 407-409 in our conclusion. However, we can see how we may have spent too much time talking about this and how it may be confusing to the reader, so we will definitely follow your recommendation and clarify this part of our paper.

To Reviewer 9:
Thank you for your kind remarks on organization and writing style.

Your concern on time complexity is certainly valid, and we agree that the algorithm can become expensive. In fact, we have been investigating improvements to the bounds we provide and since our original submission, have refined the bound in the bandit scenario.

However, to the best of our knowledge, lower bounds are not even available for traditional swap regret with oblivious adversaries ([4] provides lower bounds only for adaptive adversaries), so finding the optimal asymptotics in our situation is highly nontrivial. Still, this is definitely something we plan to investigate in the future.

To respond to your comment about the relationship between our scenario and previous results for adaptive adversaries, we discuss in section 2 lines 145-156 and in the conclusion how our scenario differs from those of an adaptive adversary because we focus on strengthening the competitor class. We also point out how, depending on the application, our scenario and result may actually be more useful than the case of an adaptive adversary (e.g. when applied to the financial markets). In light of your suggestion, we will spend some time in our paper explaining how online learning scenarios intrinsically provide two different obstacles for the learner: the learner must not only perform well against the adversary but also against his or her competitors.